# Peristomal Skin Complications in Ileostomy and Colostomy Patients: What We Need to Know from a Public Health Perspective

**DOI:** 10.3390/ijerph20010079

**Published:** 2022-12-21

**Authors:** Floriana D’Ambrosio, Ciro Pappalardo, Anna Scardigno, Ada Maida, Roberto Ricciardi, Giovanna Elisa Calabrò

**Affiliations:** 1Section of Hygiene, University Department of Life Sciences and Public Health, Università Cattolica del Sacro Cuore, 00168 Rome, Italy; 2VIHTALI (Value in Health Technology and Academy for Leadership & Innovation), Spin-Off of Università Cattolica del Sacro Cuore, 00168 Rome, Italy

**Keywords:** peristomal skin complications, PSCs, ileostomy, colostomy, ostomy surgery, burden of disease, public health

## Abstract

Background: Peristomal skin complications (PSCs) are the most common skin problems seen after ostomy surgery. They have a considerable impact on a patient’s quality of life and contribute to a higher cost of care. Methods. A systematic review was conducted, querying three databases. The analysis was performed on international studies focused on the clinical-epidemiological burden of PSCs in adult patients with ileostomy/colostomy. Results: Overall, 23 studies were considered. The main diseases associated with ostomy surgery were rectal, colon and gynecological cancers, inflammatory bowel diseases, diverticulitis, bowel obstruction and intestinal perforation. Erythema, papules, skin erosions, ulcers and vesicles were the most common PSCs for patients with an ostomy (or stoma). A PSCs incidence ranging from 36.3% to 73.4% was described. Skin complications increased length of stay (LOS) and rates of readmission within 120 days of surgery. Conclusions: PSCs data are still limited. A knowledge of their burden is essential to support health personnel and decision-makers in identifying the most appropriate responses to patients’ needs. Proper management of these complications plays a fundamental role in improving the patient’s quality of life. A multidisciplinary approach, as well as increased patient education and their empowerment, are priority measures to be implemented to foster a value-based healthcare.

## 1. Introduction

Peristomal skin complications (PSCs) are the most common complications after ostomy surgery [1]. The creation of an abdominal stoma (or ostomy) is a common procedure, performed by surgeons as part of the treatment for both benign and malignant diseases. It is a surgery fraught with complications such as necrosis, leakage, granuloma formation, retraction, stenosis, prolapsed and parastomal hernia, and also PSCs [2]. In the United States (US), there are approximately more than 750,000 persons with an ostomy and approximately 130,000 new ostomies are performed annually [3]. Instead, in Europe the available data are heterogeneous. There are approximately 20,000 ostomized people in Portugal [4], 70,000 in Italy [5], and 100,000 in Germany [5,6]. Furthermore, it is estimated that 1.5 out of every 1000 Spanish citizens has an ostomy. This number equates to 70,000 people, with over 13,000 new cases every year [7].

The main conditions requiring intestinal stoma as part of their management are colorectal cancers, inflammatory bowel diseases (IBDs), a diverticular disease with obstruction, penetrating bowel injuries, ischemic colitis, radiation injury, and fecal incontinence [8]. The underlying disease leading to ostomy surgery, the type of surgery performed (elective or emergency) and the patient characteristics are some of the main factors that favor ostomy complications [9]. These complications can be early and late, and they range from 20% to 70% rates of occurrence [9]. In particular, early complications occur within 30 days of surgery, and their incidence ranges from 3–82% [10]. Early complications include stomal ischemia/necrosis, retraction, mucocutaneous separation, and parastomal abscess, while late complications include parastomal hernia, prolapse, retraction, and varices [9]. Late complications are defined as having occurred after the physiological adjustment period that generally ranges from six to 10 weeks. Most late complications occur within the first six months after surgery but can also occur up to 15 years after the creation of an ostomy. Overall, late complication rates range from 6% to over 76% [11]. PSCs may occur at any time [12], but the incidence is highest in the first five years after surgery [13]. Furthermore, these types of complications occur more frequently in patients with ileostomy than in patients with urostomy and colostomy [13,14]. The severity of PSCs varies from mild erythema to eroded or ulcerated skin, but many different skin problems can arise in ostomized patients. These include fecal dermatitis, mechanical dermatitis, folliculitis, psoriasis, allergic contact dermatitis, peristomal pyoderma gangrenosum (PPG) and other rather uncommon conditions [15]. Indeed, the PSCs etiology is complex and multifactorial and depends on several factors, including preoperative preparation and postoperative care [3,16].

While substantial attention has been focused on the surgical complications in the published literature, little attention has been paid on PSCs [1]. However, PSCs are an important challenge for a great majority of individuals with a stoma [17]. Indeed, they can have a tremendous negative impact on health-related quality of life for the patient [18]. Furthermore, the burden for the healthcare professionals (HCPs) and the associated healthcare costs increase when PSCs become more severe [3]. The high prevalence of people living with PSCs comes with considerable economic costs for the society [13,19], and prevention and proper handling of these complications is crucial.

Therefore, the aim of this systematic review is to determine the epidemiology and clinical burden of PSCs in ileostomy and colostomy patients. The logic in this approach is that a health needs assessment is a critical step in planning patient-centered and value-based health services.

## 2. Materials and Methods

A systematic literature review was performed to evaluate the clinical-epidemiological burden of PSCs in adult patients with ileostomy and colostomy. The systematic review was conducted according to the Preferred Reporting Items for Systematic Reviews (PRISMA) [20].

### 2.1. Search Strategy

The literature search was performed by consulting three databases, namely, PubMed, Web of Science (WoS) and Scopus. The search strings were launched on 7 September 2022. The systematic review was performed from 1 January 2012. The following search string was used on PubMed:

(“peristomal”[All Fields] AND (“skin”[MeSH Terms] OR “skin”[All Fields]) AND (“complicances”[All Fields] OR “complicate”[All Fields] OR “complicated”[All Fields] OR “complicates”[All Fields] OR “complicating”[All Fields] OR “complication”[All Fields] OR “complication s”[All Fields] OR “complications”[MeSH Subheading] OR “complications”[All Fields])).

This spelling was then adapted to WoS and Scopus databases. The following filters were applied: studies on humans and in English language. The article records were entered in an Excel work sheet and screened according to the inclusion/exclusion criteria. A check for duplicates was performed; the selection was made firstly by reading titles and abstracts, and then the full texts.

### 2.2. Inclusion/Exclusion Criteria

The studies on the clinical-epidemiological burden of PSCs in adult patients with ileostomy and colostomy were considered potentially eligible. We included original articles and systematic reviews, written exclusively in English language and published as of 1 January 2012. Narrative reviews, commentary, editorials, conference presentation, and references not provided with full text, as well as studies conducted in animals or in vitro, were excluded.

### 2.3. Selection Process and Data Extraction

Four researchers (F.D., C.P., A.M., A.S.) independently screened titles and abstracts first, and full texts afterwards. Any disagreement was resolved by discussion or by the involvement of a senior researcher (G.E.C.).

Furthermore, the included studies were subjected to the snowballing process, using the bibliographic references, in order to identify additional articles that met the inclusion criteria of our review.

From the articles definitively included in the literature review, the following information was extracted: first author’s name, publication year, country, study type, sample size, characteristics of the population (age and gender), type of ostomy (ileostomy or colostomy), causes of surgery, and skin complications details (epidemiological data, PSCs type, risk factors, time-to-onset of skin problems, and use of healthcare services).

## 3. Results

The overall research in the three databases yielded a total of 549 articles. After duplicates removal, 287 articles were screened based on title and abstract. In total, 70 full-text articles were selected. Following the inclusion and exclusion criteria, the screening resulted in the final inclusion of 22 articles. One new study was included after the snowballing process [21]. Details about the study selection process are shown in Figure 1.

Of the 23 studies included in our systematic review, 11 (47.9%) were retrospective studies [3,12,13,21,22,23,24,25,26,27,28], five (21.8%) were prospective studies [29,30,31,32,33], three (13.0%) were cross-sectional studies [5,18,34], three (13.0%) were surveys [35,36,37] and one (4.3%) was a systematic review [38].

Among the 22 primary studies, 36.5% were conducted in the US (n = 8) [9,12,13,18,24,29,34,35], 18.3% in Turkey (n = 4) [22,23,27,32], 9.1% in Japan (n = 2) [25,28], 4.5% in China (n = 1) [21], 4.5% in India (n = 1) [33], 4.5% in UK (n = 1) [37], 9.1% in Sweden (n = 2) [30,31], 4.5% in Switzerland (n = 1) [26], 4.5% in Italy (n = 1) [5] and, eventually, one (4.5%) was conducted at multinational level [36].

All 23 studies [3,5,12,13,18,21,22,23,24,25,26,27,28,29,30,31,32,33,34,35,36,37,38] reported PSCs data on adult populations and, when specified, patients were predominantly male, with a mean age ranging from 47 [23] to 70 years [31]. All studies considered patients with both ileostomy and colostomy, and only 17.4% of the articles (n = 4) included patients with ileostomy only [21,23,28,33]. The main underlying diseases requiring the ostomy surgery were cancers, reported in 16 studies (69.6%) [5,12,13,21,22,23,24,25,26,28,29,30,31,32,35,38]; diverticulitis, reported in five studies (21.7%) [5,12,30,35,38]; IBDs, reported in four studies (17.4%) [12,22,30,31], and bowel obstruction and intestinal perforation, reported in two studies (8.7%) [12,22].

Among the oncological causes, colorectal cancer featured in 34.9% (n = 8) of the studies [21,24,25,28,29,31,32,38], followed by gynecological cancers (8.7%; n = 2) [24,30] and gastrointestinal ones (4.3%; n = 1) [12]. In addition, four studies (17.4%) discussed the classification of PSCs rather than providing epidemiologic data on these complications [5,18,31,34].

We report below a description of the main findings of this systematic review by distinguishing the following five thematic areas: clinical-epidemiological burden of PSCs, time of onset, assessment tools, risk factors, and hospital admission and readmission related to PSCs. The main features of each study are summarized in Table 1 and Table 2.

### 3.1. Clinical-Epidemiological Burden of PSCs

The epidemiological data on PSCs are still limited and those available are heterogeneous, both in terms of incidence and prevalence. Reasons for discrepancies in these estimates include relatively small and/or heterogeneous sample sizes, differences in the types of ostomies studied, differences in the types of complications considered and how cases were identified, and variability in assessment periods.

Our systematic review revealed that rates of PSCs incidence, following ostomy surgery, range from 36.3% [3] to 73.4% [37]. In a 2019 retrospective study, Taneja et al. [3] reported that approximately one-third of 168 subjects studied had a PSC in the 90 days following surgery, with an overall incidence of 36.3%, which was similar to the result reported by the same authors in a previous study (36.7%) [13]. Salvadalena et al. [29] reported a 63% higher incidence of PSCs within 90 days of ostomy surgery, in 43 ostomized patients. Instead, Voegeli et al. [37] reported a PSCs incidence equal to 73.4% in a population of 4.235 patients with ostomy, and Baykara et al. [22] reported a PSCs incidence equal to 48.7% in a sample of 748 ostomized patients. Data on the prevalence of PSCs are also heterogeneous. Indeed, in a Swedish study, Carlsson et al. [31] documented a prevalence rate of PSCs equal to 11% one year after ostomy surgery, in a sample of 207 patients with stoma. Instead, a higher prevalence (88%) was reported by Fellows et al. [36] in a sample of 54,000 patients involved in a multinational survey. Additionally, Lindholm et al. [30] reported a higher prevalence of PSCs after hospital discharge (45% at both three and six months, 21% at 12 months and 18% at 24 months).

PSCs were reported more frequently in patients with ileostomy than colostomy [27,31,32]. For example, Ayik et al. [27], in their study conducted on 572 patients with ostomy, reported that early PSCs occurred more frequently in patients with ileostomy (43.8%) [27]. Likewise, among the patients with PSCs enrolled in the study of Carlsson et al. [31], 6% had a colostomy and 23% had an ileostomy. Instead, Harputlu et al. [32], reported a higher frequency of skin sequelae in patients with ileostomy than those with colostomy (50% vs. 16%).

The most frequent type of PSCs reported in the literature was peristomal contact dermatitis, described in eight studies (34.8%) [12,21,24,27,32,33,34,35], which has an incidence range of 17.31% [21] to 91.7% [33].

Cressey et al. [24] listed a wide range of clinical reaction patterns related to peristomal contact dermatitis, including erythema in all the observed cases, along with erosion or even skin ulceration or vesiculation [24]. Other common PSCs were peristomal moisture-associated skin damage (PMASD) (~50.7%), defined as irritation which caused the skin to be inflamed, sore, itchy, and red [12,25,29,37], followed by maceration (~20.5%) [12], mechanical trauma (16.4%) [12,32], skin infections (e.g., fungus or folliculitis) and PPG (~1.4%) [12,31]. In addition, three studies included in our review reported other PSCs-related issues, including symptoms such as pruritus, pain, itching, burning, bleeding, and ulcers [24,25,36,37]. More details are reported in Table 2.

### 3.2. Time of onset of PSCs

More than 43% of the studies included in our systematic review addressed the time of onset of PSCs after ileostomy or colostomy surgery [3,12,13,21,25,26,27,29,32,33].

As reported by Salvadalena et al. in 2013 [29], the onset of PSCs occurred more frequently after 21–40 days post-surgery [29], although in another recent study by the same author [12], PSCs occurred 64 days after ostomy surgery (SD = 29.8 days; range: 16–132 days). Instead, Taneja et al. [3] reported an average time of onset of PSCs equal to 26.4 days after ostomy surgery; in particular, the time of onset was about 24 days for patients with ileostomy and about 27 days for those with colostomy. In another study, the same author estimated a PSCs incidence of 36.7% after 90 days after ostomy surgery with a mean time to their onset of 23.7 ± 20.5 days and, specifically, of 23.2 ± 20.8 days in patients with colostomy, and 24.2 ± 21.1 days in patients with ileostomy [13].

Furthermore, in our review it was found that among patients with early-onset PSCs, irritative dermatitis was reported in 40% of studies (n = 4) [21,27,32,33]. In both the early and late periods, peristomal irritant contact dermatitis (PICD) was the most common type of PSCs reported by Ayik et al. (31.6% and 26%, respectively) and occurred mainly within the second or third week after ostomy surgery [27]. Conversely, Nagano et al. reported the highest rate of PMASD (51.9%) in the eighth week (49–55 days) after ostomy surgery [25].

### 3.3. Classification and Assessment Tools

To date, there is important heterogeneity in the classification systems used to classify PSCs and no standardized assessment tool is available (Table 2).

Nine studies (39.1%) included in our review addressed the issue of PSCs classification, adopting or comparing various tools to assess and classify these skin complications [5,12,18,26,30,31,32,34,36].

Carlsson et al. proposed a clinical classification of PSCs that included mild erythematous-erosive skin lesions (E+) (the most frequent), severe erythematous-erosive skin lesions (E++), and skin ulcerations diagnosed as PPG [31]. Depending on the intensity and extension of the erythematous-erosive lesions, Lindholm et al. [30] also found about 4–19% of severe PSCs (E++) in a sample of 144 patients.

Menin et al. identified three main typologies of PSCs: elementary lesions, in which erythema was the most frequently reported (17.3%); ulcerative lesions, that affected more than half of the sample with erosions (32.7%) or less severe ulcers (31%); or lesions with overgrowth of tissue, reported in 12.7% cases [5].

Carbonell et al. categorized PSCs as modifying a validated scale of peristomal skin lesions and classified the severity of complications to mild or relevant [26].

Moreover, according to severity, Salvadelena et al. adopted a score scale ranging from 0 to 15 and grouped the PSCs into mild (1–3), moderate (4–6), and severe (7–15) [12]. Conversely, Harputlu et al. [32] used an international tool to describe the severity, the extent and likely cause of peristomal skin disorders among 35 patients at follow-up.

Eventually, a self-assessment tool based on PSCs visual signs was adopted by Fellows et al., classifying skin lesions from no signs of discoloration, to mild, moderate, or severe discoloration [36]. Instead, Nichols T. [34] classified peristomal skin irritation into three levels, including Level 1, corresponding to peristomal skin integrity that is intact, with no presence of irritated skin (presented by 551 of 2329 patients); Level 2, corresponding to a low-to-moderate level of reddening and irritation, with occasional but slight blistering (presented by 1029 of 2329 patients); and Level 3, corresponding to severe irritation and reddening, along with severe blistering resulting in denuded skin and ulceration (presented by 427 of 2329 patients) [34].

In another study, Nichols et al. classified skin irritation according to severity into intact (normal) skin (551 out of 2260 patients); mild to moderate skin irritation (1029/2260); and severely irritated skin (427/2260) [18].

### 3.4. Risk Factors Related to PSCs

Several risk factors, related to the onset of PSCs, were described. The impact of factors related to ostomy surgery (e.g., type of procedure, ostomy site marked, type of ostomy, length of surgery) varied from study to study [22,26,27]. Two studies reported a high rate of PSCs in patients undergoing emergency procedures and with an unmarked ostomy site before surgery [22,26].

PSCs were found to develop at a higher rate in individuals with an ileostomy [13,22,23,25,31,33,37,38]. The study of Voegeli et al. revealed that the risk of experiencing more PSCs in patients with an ileostomy was 1.9 times higher than for those with a colostomy [37].

Other individual factors, such as female gender, BMI, patient age (decreasing with age) and underlying diagnosis and comorbidities were also differently reported [3,21,26,27,28,32,33,37]. Obesity and diabetes were frequently mentioned to be predictive of PSCs [21,26,27,28]. In the study conducted by He et al., diabetes mellitus was a risk factor for early post-operative peristomal dermatitis in patients with ileostomy [21].

Moreover, two studies also found that patients undergoing post-operative chemotherapy or/and radiotherapy were more likely to experience PSCs [25,32]. Regarding the female gender, the risk of reporting a PSC was observed to be 1.35 higher in women than in men [37]. In addition, the analysis performed by He et al. [21] suggested that gender was an independent factor for peristomal dermatitis, and that females were more likely to suffer it.

### 3.5. Hospital Admissions and Readmissions Related to PSCs

Despite the limited number of studies on PSCs management, the data collected suggested that PSCs lead to the increased use of healthcare resources [3,13,28] and, consequently, higher healthcare costs.

Taneja et al. [13] reported that patients with PSCs had longer hospital stays, with an average of 21.5 days versus 13.9 days for those without these complications. Patients with PSCs were also more likely to have hospital readmissions within the 120 days following surgery (47% vs. 33% without PSCs) [13]. Furthermore, the patients with PSCs had substantially higher costs of post-surgical care than those without skin complications. Furthermore, it was estimated that the total healthcare costs over 120 days averaged USD 204,907 among patients with PSCs and USD 126,747 among those without PSCs [13]. Additionally, Maeda et al. [28] reported more hospital readmissions in patients with PSCs, while in another study, Taneja et al. [3] reported that patients with PSCs were more likely to have hospital readmissions within the 120 days (55.7% vs. 35.5% for those without complications) after ostomy surgery, with a mean length of stay equal to 11.0 days for patients with PSCs and 6.8 days for those without PSCs [3].

**Table 1 ijerph-20-00079-t001:** Main characteristics and findings on PSCs of the systematic review included in our study.

First Author, Year, [Ref.]	Study Type	N. of Included Studies	Type of Ostomy Surgery	Underlying Diseases Leading to Ostomy Surgery	Main Findings
Malik T.A.M, 2018, [38]	Systematic review	18 trials	Ileostomy and Colostomy	Colorectal cancer, diverticular disease, fecal incontinence, constipation, irritable bowel syndrome, typhoid, tuberculosis, trauma, colovesical fistula and familial adenomatous polyposis syndrome	PSCs had the highest incidence across all ostomy types at 14.0% (2.4–46.2%), followed by parastomal hernia, which occurred in 5.5% of patients (0–88.2%). PSCs were most common in patients with a loop ileostomy (median 14.0%) and loop colostomy (median 32.3%).

**Table 2 ijerph-20-00079-t002:** Main characteristics and findings of the primary studies included in our systematic review.

First Author, Year, Country, [Ref.]	Study Type	Sample Size and Population Characteristics	Type of Surgery (Ileo- or Colostomy)	Underlying Diseases	Epidemiological Data on PSCs	Type of PSCs and Related Data	Classification Tools	Risk Factors Related to PSCs	Time-to-Onset of PSCs	Hospitaladmissions/Readmissions/Other Healthcare Services Costs
Salvadalena G.D, 2013, US, [29]	Prospective study	Tot: 47 patients; M: 24 patients (51%); F: 23 patients (50%); Age: 47.6 ± 15.2 years (range 20–81 years)	Colostomy: 8 patients (17%); Ileostomy: 37 patients (79%); Urostomy: 2 patients (4%)	Ulcerative colitis; Crohn disease; Colorectal cancer; Perforated colon; Bladder cancer; Fistula; Others (Clostridium difficile colitis, indeterminate colitis, failed ileoanal pouch, familial adenomatous polyposis)	PSCs combined cumulative incidence: 63%	Moisture-associated skin damage: 14 patients; Skin infections (e.g., fungus or folliculitis): 11 patients; Erosion (excoriated; moist and bleeding): 8 patients; Erythema: 7 patients.	N.A.	N.A.	Most frequently 21–40 days after surgery ostomy	N.A.
Lindholm E, 2013, Sweden, [30]	Prospective study	Tot: 144 patients; Mean age: 67 years (range 53.5–78 years)	End colostomy: 84 patients (58%); Loop colostomy: 10 patients (7%); End ileostomy: 26 patients (18%); Loop ileostomy: 24 patients (17%).	Diverticulitis: 41 patients (28%); Gynecological cancer: 28 patients (19%); Rectal cancer: 19 patients (14%); Colon cancer: 18 patients (13%); IBD: 15 patients (10%); Other types of cancers, fistulas, or sphincter rupture after delivery: 23 patients (16%).	PSCs Prevalence: at 3–6 months: 45%; at 12 months: 21%; at 24 months: 18%.	N.A.	Severe peristomal skin problems (classified as E++): 4–19%	N.A.	On ward: 9 patients (6.5%); After 2 weeks: 22 patients (19.3%); After 3 months: 14 patients (13.6%); After 6 months: 12 patients (16.9); After 1 year: 3 patients (5.3%); After 2 year: 1 patient (3.8%).	N.A.
Baykara Z.G, 2014, Turkey, [22]	Retrospective study	Tot: 748 patients; M: 408 patients (54.5%); F: 340 patients (45.5%); Mean age: 56.60 ± 16.73 years.	Ileostomy: 363 patients (48.5%); Colostomy: 354 patients (47.3%)	Cancer: 545 patients (72.9%); IBD: 58 patients (7.8%); Bowel obstruction: 35 patients (4.7%); Injuries: 34 patients (4.5%); Intestinal perforation: 14 patients (1.9%); Fistula: 13 patients (1.7%) Familial adenomatous polyposis: 11 patients (1.5%); Anorectal malformation: 6 patients (0.8%); Mesenteric ischemia: 6 patients (0.8%); Sigmoid volvulus: 6 patients (0.8%); Intra-abdominal abscess: 4 patients (0.5%); Other: 16 patients (2.1%)	PSCs rate by type of surgery Emergency: 43 patients (19.5%); Planned: 93 patients (17.6%)	Peristomal skin problems: 136 (48.7%) Maceration: 2 (0.7%) Allergy: 1 (0.4%)	N.A.	Unplanned/ Emergency ostomy procedure; Multiple ostomies; Type of ostomy.	N.A.	N.A.
Sarkut P, 2015, Turkey, [23]	Retrospective study	Tot: 141 patients; M: 95 patients (67%); F: 46 patients (33%); Mean age: 47 years (range: 17–67 years)	End ileostomy 43%; Loop ileostomy 46%; Double-barrel ileostomy 11%.	Benign causes: 48%; Malign causes: 52%	N.A.	Maceration in the peristomal skin: 10 patients	N.A.	N.A.	N.A.	N.A.
Carlsson E, 2016, Sweden, [31]	Prospective Study	Tot: 207 patients; F: 53% Mean age: 70 years (range 19–94 years); Elective surgery: 74%	Colostomy: 146 patients (71%); End ileostomy: 54 patients (26%); Loop ileostomy: 7 patients (3%)	Colorectal cancer: 62%; IBD: 19%	PSCs prevalence: 23 patients (11%); PSCs with colostomy: 9 patients (6%); PSCs with an end or loop ileostomy: 14 patients (23%)	N.A.	Erythematous-erosive skin lesions (E+): 16 patients; Erythematous-erosive skin lesions (E++): 5 patients; Ulcerations (pyoderma gangrenosum): 2 patients	N.A.	N.A.	N.A.
Cressey B.D, 2017, USA, [24]	Retrospective study	Tot: 18 patients; M: 11 patients; F: 7 patients; Mean age: 60.4 years (range: 35–87 years)	Colostomy: 9 patients; Ileostomy: 3 patients; Ileal conduit diversions: 6 patients	Cancer (colorectal: 7 patients; genitourinary: 6 patients; ovarian: 2 patients)	N.A.	Peristomal contact dermatitis: 12 patients; Erythematous extending out from the stoma: 18 patients; Erosion: 1 patient; Ulceration: 1 patient.	N.A.	N.A.	N.A.	N.A.
Taneja C, 2017, US, [13]	Retrospective study	Tot: 128 patients; M: 67 patients (52.3%); F: 61 patients (47.7%); Mean Age: 60.6 ± 15.6 years	Colostomy: 51 patients (40%); Ileostomy: 64 patients (50%)	N.A.	PSCs incidence: 36.7% within 90 days following surgery [35.3% (n = 18) with colostomies; 43.8% (n = 28) with ileostomies].	N.A.	N.A.	N.A.	Average time from surgery to first notation of PSCs: 23.7 ± 20.5 days; Colostomy: 23.2 ± 20.8 days; Ileostomy: 24.2 ± 21.1 days.	The mean length of stay for the index admission: 21.5 days for patients with PSCs vs. 13.9 days for all other patients. Readmissions: 22 patients (46.8%) with PSCs vs. 27 patients (33.3%) without PSCs; Readmissions for PSCs with colostomy: 8 patients (44.4%); Readmissions for PSCs with ileostomy: 14 patients (50.0%); Mean number of outpatient care visits: 11.4 (6.2%); Outpatient care visits for PSCs with colostomy: 13.4 (6.3%); Outpatient care visits for PSCs with ileostomy: 10.4 (5.7%); Mean number of home care visits: Home care visits for PSCs with colostomy: 9.2 (4.4%); -Home care visits for PSCs with ileostomy: 8.7 (5.0%).
Harputlu D.U, 2018, Turkey, [32]	Prospective study	Tot: 35 patients; F: 22 patients (62.9%); Mean age: 57.45 ± 14.70 years; Intervention group (home care visits): 18 patients; Control group (outpatient/ clinic care): 17 patients	Ileostomy: 21 patients (60.0%); Permanent ostomy: 18 patients (51.4%); Unspecified ostomy: 8 patients (22.9%)	No chronic disease: 20 patients (57.1%); Rectal cancer: 14 patients (40%); Unspecified cause: 1 patient (2.9%)	N.A.	PSCs in Intervention group (Home nursing care): Irritant dermatitis: 12 patients (66.7%); 16.7% with a colostomy and 50% with an ileostomy. Mechanical trauma: 3 patients (16.7%); Allergic dermatitis: 2 patients (11.1%); Both allergic and irritant dermatitis: 1 patient (5.6%) PSCs in Control group (Outpatient/clinic care): Irritant dermatitis: 14 patients (82.4%); 23.5% with a colostomy and 58.8% with an ileostomy. Mechanical trauma: 2 patients (11.8%); Allergic dermatitis: 1 patient (5.9%)	Application of the OST to describe the severity, extent, and likely cause of a peristomal skin disorder	Diabetes mellitus; Immobility: Chemotherapy or/and radiotherapy	Irritant dermatitis in intervention group: appearance in the early postoperative period (0 to 29 days) in 2 patients with colostomy and 3 with ileostomy; Irritant dermatitis in control group: appearance in the early postoperative period (0 to 29 days) in 3 patients with ileostomy	N.A.
Nichols TR, 2018 US, [34]	Cross-sectional study	M: 1230 patients; Mean age: 53.5 years (range 65.1 ±12.6 years); F: 1070 patients; Mean age: 46.5 years (range 61.8 ±13.4 years)	Ileostomy: 1031 patients (44.3%); Colostomy: 920 patients (39.5%); Urostomy: 308 patients (13.2%); Multiple types: 33 patients (1.4%)	N.A.	N.A.	Peristomal skin irritation	Self-report assessment: Level 1: peristomal skin integrity, no presence of irritated skin (n = 551 patients); Level 2: low to moderate level of reddening and irritation, occasional but slight blistering (n = 1029 patients); Level 3: severe irritation and reddening with severe blistering, denuded skin and ulceration (n = 427 patients).	N.A.	N.A.	N.A.
Nichols & Inglese, 2018 US, [18]	Cross-sectional study	M: 1230 patients (53.48%); Mean age: 65.12 ± 12.62 years; F: 1030 patients (46.52%); Mean age: 61.77 ± 13.43 years	Colostomy: 920 patients (39.50%); Ileostomy: 1031 patients (44.27%); Urostomy: 308 patients (13.22%); Multiple stomas: 33 patients (1.42%); Unknown: 37 patients (1.59%)	N.A.	N.A.	Skin irritation	Self-report assessment: Intact (normal) skin: 551 patients; Mild to moderate skin irritation: 1029 patients; Severely irritated skin: 427 patients	N.A.	N.A.	N.A.
Menin G, 2019, Italy, [5]	Cross-sectional study	Tot: 110 patients; M: 57 patients (51.8%); F: 53 patients (48.2%); Mean age: 69 years (range 19–90 years)	Ileostomy 47.3%; Colostomy 38.2%; Other types of procedures 14.5%.	Cancer: 58 patients (52.7%); Chronic intestinal inflammatory: 10 patients (9.1%); Diverticulitis: 9 patients (8.2%); Others: 33 patients 30%	N.A.	Erythema: 19 patients (17.3%); Papules: 2 patients (1.8%); Pustules: 2 patients (1.8%); Vesicles: 2 patients (1.8%); Bubbles (>0.5 mm): 1 patients (0.9%); Overgrowth of tissue: 14 patients (12.7%); Erosion: 36 patients (32.7%); Healing ulcer: 5 patients (4.6%); Mixed ulcer: 21 patients (19.1%); Worsening ulcer: 8 patients (7.3%)	The Peristomal Lesion Scale (PLS) vs. SACS Instruments: PSCs according to PLS: Elementary (Erythema, Papules, Pustules, Vesicles, Bubbles (>0.5 mm): 26 patients; Overgrowth of tissue: 14 patients; Ulcerative (Erosion, Healing ulcer, Mixed ulcer, Worsening ulcer): 70 patients. -PSCs according to SACS classification *: L1: 22 patients (20%); L2: 39 patients (35.5%); L3: 22 patients (20%); L4: 11 patients (10%); LX: 15 patients (14.5%)	-	-	-
Taneja C, 2019, US, [3]	Retrospective study	Tot: 168 patients; M: 78 patients (46.4%); F: 90 patients (53.6%); Mean age: 63.9 years	Colostomy: 108 patients (64.3%); Ileostomy: 40 patients (23.8%)	N.A.	PSCs incidence: 36.3%	N.A.	N.A.	N.A.	PSCs within 90 days of ostomy surgery: 36.3% (ileostomy, 47.5%; colostomy, 36.1%); Mean time from surgery to first notation of a PSC: 26.4 days; Ileostomy group: 24.1 days; Colostomy groups: 27.2 days	Patients with PSCs were more likely to be readmitted to hospital by day 120 (55.7% vs. 35.5% for those without PSCs); Mean length of stay for PSCs patients readmitted to hospital: 11.0 days vs. 6.8 days for those without PSCs; Mean number of outpatient care visits: Colostomy: 7.4 (6.3%); Ileostomy: 5.7 (2.0%), Mean total PSCs healthcare cost over 120 days per patients: USD 58,329 vs. USD 50,298 for those without PSCs
Nagano M, 2019, Japan, [25]	Retrospective study	Tot: 89 patients; M: 58 patients (65.2%); F: 31 patients (34.8%); Mean age: 65 years	Ileostomy: 52 patients (58.4%); Colostomy: 37 patients (41.6%)	Colorectal cancer	N.A.	MASD	N.A.	Ileostomies; Temporary stomas; Chemotherapy	8 week after ostomy surgery: 51.9% of MASD	N.A.
Voegeli D., 2020, UK, [37]	Multinational survey	Tot: 4235 patients; M: 55% F: 45%	Colostomy: 43%; Ileostomy: 38%; More than one: 2%	N.A.	PSCs self-reported incidence: 73.4%	Itching: 67%; Bleeding: 45%; Discoloration: 38%; Burning: 32%; Moisture from damage: 28%; Pain: 21%; Wounds:11%; Tissue overgrowth: 7%	N.A.	Higher risk of PSCs after ileostomy surgery: 1.9 higher than in those with colostomy; 1.5 times higher risk of PSCs in the first 2 years after ostomy surgery; Greater risk in women: 1.35 more than in men.	N.A.	N.A.
Carbonell B.B, 2020, Switzerland, [26]	Retrospective study	Tot: 111 patients; M: 64 patients (58%); F: 47 patients (42%); Mean age: 67.61 ± 15	Colostomy: 40 patients (36%); Ileostomy: 71 patients (64%)	Malignant disease: 72 patients (65%); Benign disease: 39 patients (35%)	PSCs rate: 73%	Mild complications: Hyperemic lesion: 6 patients (5%); Erosive lesion: 25 patients (23%); Suture Fissure: 9 patients (8%); Relevant complications: Ulcerative lesion: 6 patients (5%); Muco-cutaneous separation: 57 patients (51%); Abscess: 3 patients (3%); Retraction: 5 patients (5%); Necrosis: 2 patients (2%)	SACS classification	Predictors for persistence of peristomal complications at 30 postoperative days: ASA score III/IV; urgent surgery	Early peristomal complications are common, usually mild. They are most likely to persist beyond 30 days in patients operated as emergencies and with an ASA score of III-IV	N.A.
Salvadalena G, 2020, US, [12]	Retrospective study	Tot: 73 patients; M: 44 patients; F: 29 patients; Mean age: 56.2 ± 14.2 years	Colostomy: 35 patients (48.0%); Ileostomy: 33 patients (45.2%)	Bowel cancer: 36 patients (49.3%); Bladder cancer: 4 patients (5.5%); Crohn’s disease: 4 patients (5.5%); Ulcerative colitis: 8 patients (11.0%); Diverticulitis: 11 patients (15.1%); Familial polyposis: 1 patients (1.4%); Intestinal obstruction and/or perforation: 5 patients (6.8%); Other: 10 patients (13.7%)	N.A.	Irritant dermatitis: 37 patients (50.7%); Maceration: 15 patients (20.5%); Mechanical trauma: 12 patients (16.4%); Folliculitis: 3 patients (4.1%); Pyoderma gangrenosum, fungal rash, skin infection: 1 patient (1.4%)	Severities were grouped into mild, moderate, and severe, using a range score (0–15).	Stoma duration and/or peristomal creases; Increased risk of PSCs for every 1-week increase in ostomy duration.	64 days after undergoing ostomy surgery	N.A.
Ayik C, 2020, Turkey, [27]	Retrospective study	Tot: 572 patients; M: 302 patients; F: 270 patients; Mean age: 59.15 ± 13.86 years	End colostomy: 253 patients (44.2%); Loop colostomy: 40 patients (7%); End ileostomy: 151 patients (26.4%); Loop ileostomy: 128 patients (22.4%)	N.A.	N.A.	Early PSCs: PICD: 181 patients (31.6%); - Late PSCs: PICD: 149 patients (26%)	N.A.	BMI > 24.9 kg/m^2^; Temporary ostomy; Ileostomy	Early period (<30 days after surgery): 56.5% of complications; -Late period (>30 days after surgery): 36.2% of complications	N.A.
Singh N, 2021, India, [33]	Prospective study	Tot: 36 patients; M: 28 patients; F: 8 patients; Age < 30: 19 patients; Age > 30: 17 patients	End Ileostomy: 13 patients; Loop Ileostomy: 23 patients	N.A.	N.A.	Peristomal irritation: 33 patients (91.7 %); Skin escoriation: 24 patients (66.7 %)	N.A.	N.A.	Early period (<30 days after surgery)	N.A.
Maeda S, 2021, Japan, [28]	Retrospective study	Tot: 185 patients; M: 131 patients (70.8%); F: 54 patients (29.2%); Mean age: 62 years (range: 27–83 years)	Loop Ileostomy: 185 patients (100%)	Rectal malignancies Adenocarcinoma: 174 patients (94.1%)	Skin disorders: 62 patients (33.5%)	N.A.	N.A.	Higher BMI (≥25.0 kg/m^2^); Lower ostomy height (<20 mm)	N.A.	Readmissions: 3 patients
He D, 2021, China, [21]	Retrospective study	Tot: 491 patients; M: 217 patients (65.96%); F: 112 patients (34.04%); Age ≤ 60: 171 patients (51.98%); Age > 60: 158 patients (48.02%)	Ileostomy	Colorectal cancer	N.A.	Peristomal dermatitis: 85 patients (17.31%)	N.A.	Diabetes; Female gender	Within one month after ileostomy.	N.A.
Fellows J, 2021, Multinational, [36]	Multinational survey	Tot: 5187 patients; M: 56%; F: 44%; More than 18 years of age.	Colostomy: 51%; Ileostomy: 33%	N.A.	PSCs rate: 88%	PSC with >1 or 1 symptoms/signs (e.g., pain, itching, burning): 78%	Self-report assessment: Peristomal skin with mild discoloration: 32%; Peristomal skin with medium discoloration: 16%; Peristomal skin with severe discoloration: 4%	N.A.	N.A.	N.A.
Pittmann J, 2022, US, [35]	Web-based survey	Tot: 202 patients; M: 144 patients; F: 46 patients; Mean age: 54.91 ± 14.52 years	Ileostomy: 89 patients (45.41%); Colostomy: 50 patients (25.51%); Multiple stomies: 13 patients (6.63%)	Cancer: 81 patients (46.55%); Ulcerative colitis: 39 patients (22.41%); Crohn disease: 23 patients (13.22%); Diverticulitis: 7 patients (4.02%); Trauma: 2 patients (1.15%); Other: 22 patients (12.64%)	N.A.	Peristomal skin irritation: 78% (n =135/173)	N.A.	N.A.	N.A.	N.A.

PSCs: Peristomal skin complications; N.A.: Not Available; IBD: Inflammatory Bowel Disease; OST: Ostomy Skin Tool; MASD: Peristomal Moisture-Associated Skin Damage; PICD (peristomal irritant contact dermatitis). * The SACS classification considers five types of lesion: (L1) hyperemic (peristomal erythema without loss of substance); (L2) erosive (open lesion with loss of substance, not extending into subcutaneous tissue); (L3) ulcerative (open lesion extending into subcutaneous tissue); (L4) ulcerative (full-thickness skin loss with dead tissue, fibrinous/necrotic lesion); and (LX) proliferative (abnormal growths present, i.e. hyperplasia, granulomas, neoplasia, and oxalate deposit).

## 4. Discussion

Our systematic review summarizes the currently available evidence on the PSCs burden in ileostomy and colostomy patients. The results of our review showed that PSCs are still not well investigated complications in ostomy patients. In fact, we included only 22 primary international studies on these skin complications in our review. In addition, most of these primary studies were conducted in the US and Turkey, where about 130,000 new ostomy surgeries and 8200 ileostomies/colostomies are performed annually, respectively [3,16].

However, our literature data confirmed a relevant clinical-epidemiological burden of PSCs worldwide [19].

The available data are heterogeneous, both in terms of incidence and prevalence of PSCs. In fact, our systematic review revealed rates of PSCs incidence following an ostomy surgery range from 36.3% [3] to 73.4% [37]. Similarly, the PSCs prevalence ranged from 11% to 88% [31,36]. The reasons for the discrepancies in these estimates include relatively small and/or heterogeneous sample sizes, differences in the types of ostomies studied, differences in the types of complications considered and the how cases were identified, variability in assessment/classification of the skin problems and in the time of onset of the PSCs considered in the different studies.

PSCs were reported more frequently in patients with ileostomy than colostomy [27,31,32], and in male adults with a mean age ranged from 47 [23] to 70 years [31].

A comparison of different stoma types showed that patients with an ileostomy had a 25–43% risk of developing PSCs, while those with colostomies had a 7–20% risk [14].

The main underlying diseases requiring ostomy surgery were cancers—especially colorectal cancer, gynecological cancers and gastrointestinal ones—IBDs, bowel obstruction and intestinal perforation.

The most common PSCs reported in the literature were peristomal contact dermatitis, with an incidence range from 17.31% [21] to 91.7% [33]. Other common PSCs were PMASD (~50.7%), followed by maceration (~20.5%), mechanical trauma (16.4%), skin infections (e.g., fungus or folliculitis) and PPG (~1.4%). Furthermore, other skin damages occurred which were as frequently as erythema, escorations and erosions, or even skin ulcerations and vesiculations [24].

The PSCs onset occurred more frequently, 21–40 days after surgery [29]. The average time of onset of PSCs after ostomy surgery was 26.4 days [3]; in particular, it was about 24 days for patients with ileostomy and about 27 days for those with colostomy [3].

The PSCs etiology is complex and multifactorial and depends on several factors, including peristomal moisture-associated skin damage caused by prolonged exposure to effluent from the ostomy, mechanical skin injury, bacterial or fungal infections, and hypersensitivity or allergy to ostomy products [3]. The development of PSCs is also influenced by the type of surgery, surgical technique, preoperative preparation, postoperative care, and general health status of the patients, with a heightened risk found among those suffering from obesity or diabetes [28].

Our data also suggested that PSCs lead to increased use of healthcare resources [3,13,28]. Consequently, there are higher healthcare costs, associated with longer hospital stays, higher hospital readmission rates, and higher numbers of clinic and emergency room visits, compared to patients without skin complications [3,13,27].

In addition to the clinical burden of these complications, it is equally important to consider the economic and social burden associated with PSCs, their costs to the health system, to patients and their caregivers, and their negative impact on patient quality of life [1,39].

Although the cost of ostomy care is difficult to estimate due to high variability across countries and the scarcity of data [19], an approximate additional cost of USD 8000 was reported for hospital readmissions, outpatient visits, and treatment costs in patients with PSCs [3]. Furthermore, other studies not included in our systematic review—because they mainly focused on the costs related to PSCs and not on the related epidemiological data-, also reported important information on the economic impact of skin complications in ostomized patients. For example, Martins et al. [40] reported that the amount of an average PSCs treatment episode (assumed to last 7 weeks) ranged from GBP 106.29 (approximately US USD 133) in those deemed mild to GBP 618.69 (approximately USD 776) for those deemed severe. Meisner et al. [1] also reported an increased cost of EUR 263 (vs. EUR 215 for persons without skin complications) during a seven-week treatment period. Consequently, it is essential to prevent these types of complications and to treat them at an early stage, both to ensure a better quality of life for patients and to reduce the economic burden associated with these skin complications [18].

Living with an ostomy can also be difficult, and ostomized patients generally reduce their social interactivity and begin to exhibit disorders such as anxiety or even depression [40,41,42]. This results in a loss of confidence in their social and family relationships and in their ability to return to their normal daily activities [43,44].

Therefore, considering the health, economic and social implications arising from the PSCs, their prevention, early identification, and appropriate treatment and management are crucial to improving a patient’s health and quality of life. While several studies focus on the epidemiological data of PSCs, few have focused to date on the best practices and recommendations useful for preserving peristomal skin health [45].

Nurses, and above all specialized professional figures such as the stomal therapists, play a fundamental role in the correct planning of the pre- and post-operative ostomy care management, providing advanced and tailored assistance to patients in different healthcare settings [46]. While ostomy nurses are often the first line of management, dermatologists are involved in the care of ostomy patients with complex or persistent PSCs [47]. However, a multidisciplinary approach, also undertaken in the pre-operative phase and performed with the involvement of the surgeon, the dermatologist, and the ostomy nurse is critical in order to understand the ostomy apparatus and the possible peristomal skin conditions that may arise in the post-operative phase [47]. Therefore, properly trained professionals are needed for the management of PSCs, and the implementation of standardized protocols and specific care pathways is crucial to mitigating the incidence of these common complications in ostomy patients [48,49]. At the same time, patient education interventions should be provided to support progressive self-care training performed by patients and their caregivers, who are also active participants that monitor and care for peristomal complications. In fact, individuals living with PSCs may not recognize the early signs of altered skin integrity as an issue and may not seek the advice of a healthcare professional until the problem worsens [50].

For these reasons, the prevention, early identification and treatment of PSCs are a challenge for HCPs [41,50,51] and require a multidisciplinary approach and greater patient involvement and awareness. These actions are perfectly in line with a value-based healthcare approach [48,51,52] to be offered also to patients with stoma.

Eventually, in the current context characterized by disruptive innovation, industry should also invest in the production of ostomy devices capable of reducing the risk of infections and PSCs. All stakeholders involved in the health field (health authorities, health institutions, HCPs, and industries) should work together to ensure that patients with ostomies have better answers to their health needs.

Despite the useful findings, several limitations should be considered for our study. Only articles published in English until 7 September 2022 were included, which might have led to the failure to identify all the available evidence on the clinical-epidemiological burden of skin complications in ileostomy/colostomy patients. Moreover, selection bias could not be completely ruled out, even though the screening process was performed rigorously and according to the PRISMA statement [53]. Furthermore, a quality assessment of the included studies was not performed, and we could not assess the methodological correctness of the included articles. However, in our opinion, this does not prejudice our work, as we wanted to provide an overview of the evidence on the PSCs burden in adult patients with ileostomies/colostomies without addressing the robustness of the methods used to do so. Therefore, we believe our review provides a valuable insight into the epidemiological burden and impact of PSCs internationally. However, further studies are needed to investigate the real burden of PSCs related to ileostomies and colostomies and the methods to minimize their risks, in order to prevent these lengthy, debilitating and costly complications.

## 5. Conclusions

The data on PSCs are limited and these complications are still underestimated. This is not only because of the insufficient findings in the literature, but also because the problem is overlooked by the patients themselves.

Indeed, patients with a skin disorder do not always seek professional healthcare, do nothing if a skin complication does occur or manage themselves using a skin barrier product. Therefore, estimating the PSCs burden could support HCPs called upon to identify the most appropriate responses to patients’ health needs. The management of these complications plays a critical role in improving patients’ quality of life and a multidisciplinary professional management—with the active involvement of stoma-therapists, surgeons, and dermatologists— is needed, as are greater patient education and empowerment.

Eventually, increased evidence-based knowledge could guide the development of shared health policies and best practices, as well as support a value-based decision-making process, in order to adequately address the health needs of ostomized patients.

## Figures and Tables

**Figure 1 ijerph-20-00079-f001:**
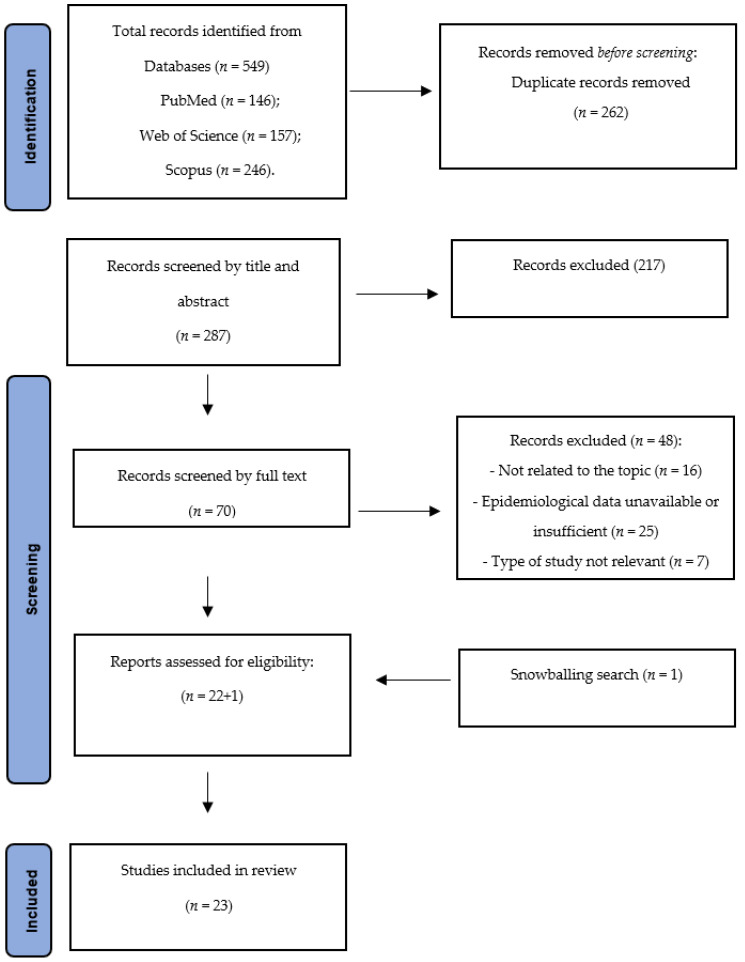
PRISMA statement flow diagram.

## Data Availability

Not applicable.

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
