# Peer review of "Peristomal Skin Complications in Ileostomy and Colostomy Patients: What We Need to Know from a Public Health Perspective"

_ijerph, 2022, doi:10.3390/ijerph20010079_

Round 1

Reviewer 1 Report

Thank you for your submission. It is an interesting article from a public health perspective, however, I do have some recommendations as follows:

1. Extensive editing of English is required: there are too many paragraphs, especially in the discussion section. Please consolidate the main points and limit discussion section to 4-5 paragraphs.

2. This article is from a public health perspective, therefore I would recommend that statistics be added into the discussions to provide better understanding to readers. 

For example: , Martins et al. [41] reported that the amount of an 425 average PSCs treatment episode (assumed to last 7 weeks) ranged from ₤106.29 (approxi- 426 mately US $133) in those deemed mild to ₤618.69 (approximately US $776) for those 427 deemed severe.

However, this reference from Martin et al. was not found in the table summary. Therefore, I would suggest to cite articles from your summary to support your public health recommendations.

Author Response

Dear Reviewer, 

We would like to thank you for the opportunity to resubmit our work. We have amended the paper according to the received suggestions and we hope that it now appears improved.
In the attached file, point-to-point answers are provided.

Best regards,

GE Calabrò

Reviewer 2 Report

The paper is interesting, well written, but duplication is found in several plases of the paper (lines 75 - 83,  348-358, 367-377, 400-412 abstract, introduction section and discussion)  and there are several writing mistakes. 

Author Response

(The authors gave the same response as above.)

Reviewer 3 Report

This is systematic review on the epidemiological burden and impact of peristomal skin complications after ostomy surgery. The authors have done a methodical and thorough work going through the literature. However, I do have some questions and comments.

Major suggestions and questions:

My main concern is that I find it difficult to understand how the authors have defined the PCS incidence and PCS prevalence. In the results (line 201-217) only four studies are sited as sources for incidence and only three as sources for prevalence. Are those really the only ones? It is stated that this review revealed an incidence of PSCs ranging between 11% and 72% (line 202-203). It is also stated that the severity of PSCs varies from mild erythema to eroded or ulcerated skin (line 56) and reference 32 had a peristomal contact dermatitis incidence of 91.7% (line 380). Do you not consider peristomal contact dermatitis to be a PCS? Also, how can the conclusion that up to 80% of individuals with an ostomy will experience PSC be drawn from a consensus statement (line 410) and how does that relate to the incidence described above.

In the results, it is stated that the PSC incidence in reference 8 was 37% (line 207) and the same is stated in the table, yet in the discussion it is stated that they reported an incidence rate between 10 and 70% (line 369). How can this difference be explained?

Minor suggestions and comments:

I suggest to reference the three articles instead of stating 3/23 in line 310. This would give more information since it is already made clear that it is a total of 23 articles included in this review. Furthermore, the results from two of these three studies are outlined but not the third, why is this?

The word “instead” (line 221 and 375) gives the impression that these results are in contrast to the previous study but in my mind, they point towards the same conclusion.

I would describe it as a risk of developing PSC rather than a chance (line 363-364).

Author Response

(The authors gave the same response as above.)

Round 2

Reviewer 2 Report

The paper is interesting and well-written, and previous mistakes are corrected.